# Biovacc-19: A Candidate Vaccine for Covid-19 (SARS-CoV-2) Developed from Analysis of its General Method of Action for Infectivity

Birger Sørensen[1]*, Andres Susrud[1] and Angus George Dalgleish[2]

[1]Immunor AS, Oslo, Norway and [2]Department of Oncology, St. George's Institute of Infection and Immunity, University of London, London, United Kingdom

Vaccine; SARS-CoV-2; Peptide; Epitope; Charge; Co-receptor

**Author for correspondence:**
*Correspondence to: B. Sørensen,
E-mail: birger.sorensen@immunor.com

**Abstract**

This study presents the background, rationale and method of action of Biovacc-19, a candidate vaccine for corona virus disease 2019 (Covid-19), now in advanced preclinical development, which has already passed the first acute toxicity testing. Unlike conventionally developed vaccines, Biovacc-19's method of operation is upon nonhuman-like (NHL) epitopes in 21.6% of the composition of severe acute respiratory syndrome coronavirus 2 (SARS-CoV-2)'s spike protein, which displays distinct distributed charge including the presence of a charged furin-like cleavage site. The logic of the design of the vaccine is explained, which starts with empirical analysis of the aetiology of SARS-CoV-2. Mistaken assumptions about SARS-CoV-2's aetiology risk creating ineffective or actively harmful vaccines, including the risk of antibody-dependent enhancement. Such problems in vaccine design are illustrated from past experience in the human immunodeficiency viruses domain. We propose that the dual effect general method of action of this chimeric virus's spike, including receptor binding domain, includes membrane components other than the angiotensin-converting enzyme 2 receptor, which explains clinical evidence of its infectivity and pathogenicity. We show the nonreceptor dependent phagocytic general method of action to be specifically related to cumulative charge from insertions placed on the SARS-CoV-2 spike surface in positions to bind efficiently by salt bridge formations; and from blasting the spike we display the NHL epitopes from which Biovacc-19 has been down-selected.

## Design methodology and parameters

Although no other corona virus disease 2019 (Covid-19) vaccine design programme appears to follow this methodology, we believe, from experience, that successful vaccine design logically starts with a thorough understanding of the aetiology of the target virus which appears in this case to be quite singular. In consequence of our researches and therefore unlike conventionally developed vaccines, Biovacc-19's method of operation is solely upon nonhuman-like (NHL) epitopes which are 21.6% of the composition of this coronavirus's spike protein. The spike displays distinct distributed charge including a charged furin-like cleavage site. Following principles previously employed to design a therapeutic human immunodeficiency viruses (HIV) therapeutic vaccine (Vacc4x), we have therefore first examined and publish here sequences and alignments of the severe acute respiratory syndrome coronavirus 2 (SARS-CoV-2) spike protein looking for unique properties of this virus that can be exploited for successful epitope presentation. We have also reviewed, and publish here, relevant cryo electron microscopy results, and structure and function relevant Cys–Cys loops, to discover what is – and is not – revealed at amino-acid level analysis of sequences by previous authors and to elicit the general mode of action for infectivity of this virus. This methodology has two parts.

In order to formulate this fact-based account of its general mode of action, first we present an explanatory model for the difference between SARS-CoV and SARS-CoV-2 supported by substantial physical/chemical data from different classes of convergent sources. Three-dimensional (3D) models and sequence analyses have been used to reveal properties associated with specific amino acids/clusters of amino acids of both spike and co-receptors. Second, we propose a theoretical explanation which future experimental model systems will be able to study to elucidate further structural details as well as to provide evidence for detailed mechanistic explanations.

These data reveal the biological structure of SARS-CoV-2 spike and confirm that accumulated charge from inserts and salt bridges are in surface positions capable of binding with cell membrane components other than the angiotensin-converting enzyme 2 (ACE2) receptor. We have also looked at the naked coronavirus spike protein as a concept for the basis of a vaccine, which we have rejected because of high risk of contamination with human-like (HL) epitopes.

Analysis of the spike protein of SARS-CoV-2 shows 78.4% similarity with HL epitopes. For the avoidance of confusion, a standard protein blast searches for functionalities and homologies to other proteins. However, antibodies can only recognize 5–6 amino acids and therefore a 6-amino

acid rolling window search for antibody epitopes was performed. A search so tailored to match against all human known proteins will give a 78.4% human similarity to the SARS-CoV-2 spike protein, that is if all epitopes on the 1,255-amino acid long SARS-CoV-2 spike protein can be used by antibodies then there will be 983 antibody binding sites which also could bind to epitopes on human proteins. This is what we did and found.

We were in the minority of vaccine designers with regard to HIV vaccine development, having concluded that a vaccine based on the envelope gp120 would not be effective. We proposed instead using the unique gag proteins as the basis of the Vacc4x vaccine which has been shown to induce robust immune responses and reduce the HIV viral load in several multicentre studies (Pollard *et al.*, 2014; Huang *et al.*, 2016; Huang *et al.*, 2017; Huang *et al.*, 2018). It is 36 years since the world was promised an HIV vaccine that would be ready in 18 months. We correctly predicted the failure of all three major HIV/AIDS vaccines over those years, and specifically the danger of poor immune responses to conserved human-like domains and antibody-enhanced infectivity to high mutating domains. Earlier this year, the latest South African trial was terminated due to futility in preventing HIV transmission (UNAIDS, 2020). From our past HIV experience, we therefore observe that in the present context, any vaccine design based on the whole spike protein of SARS-CoV-2 may not be immunogenic due its high human similarity compared to a vaccine with specifically selected NHL epitopes, such as Biovacc-19 does – and is.

Covid-19 candidate vaccines designed without appreciating these problems may run similar risks to those experienced with HIV vaccines that failed to show protection. The possibility of inducing autoimmune responses or antibody-dependent enhancements, needs to be carefully guarded against because there is published evidence that an HIV candidate vaccine has actually enhanced infectivity (Duerr *et al.*, 2012): 'Vaccinations were halted; participants were unblinded. In *post hoc* analyses, more HIV infections occurred in vaccines *versus* placebo recipients in men who had Ad5-neutralizing antibodies and/or were uncircumcised. Follow-up was extended to assess relative risk of HIV acquisition in vaccines *versus* placebo recipients over time'. Such antibody-dependent enhancement (ADE) has been observed for coronaviruses in animal models, allowing them to enter cells expressing FcγR. ADE is not fully understood: however, it is suggested that antibody-dependent enhancements may come as a result of amino acid variability and antigenic drift (Negro *et al.*, 2020; Ricke *et al.*, 2020).

## Adjuvants are not secondary considerations

Conventional vaccine methodologies tend to deal sequentially with the choice of adjuvant after the primary design work has been achieved. In contrast, we believe that the two aspects of design are indissoluble and that adjuvant choice is *ab initio* an essential aspect of a successful vaccine design. That is because it has been observed that with the right adjuvant there can be a valuable inverse correlation with infectivity, with morbidity and with fatality in published cohorts (Lam *et al.*, 2017).

Most adjuvants have a strong T-Helper 2 (TH-2) bias in order to achieve a good neutralizing antibody response. Given the results of our research into the SARS-CoV-2 aetiology, we posit that an adjuvant is required that specifically activates innate and cell mediated immunity which will give the necessary enhancement in TH-2 response to the peptide specific epitopes.

It has been known for several years that Bacillus Calmette–Guérin (BCG) can enhance a TH-1 response and can be used as an adjuvant for cell-based vaccines such as the melanoma cell-based Cancervax pioneered by Donald Morton (Faries *et al.*, 2017). However, it cannot be used repeatedly because it will induce a nonspecific humoral response that will enhance cancer progression. In an attempt to overcome this limitation, Standford and Rook developed a programme for improved agents that could replace BCG in the control of Tuberculosis and noted that heat killed *Mycobacterium vaccae* had the ability to enhance TH-1 responses and suppress the humoral response, making it a therapeutic agent in its own right (Bourinbaiar *et al.*, 2019). Further research showed that *Mycobacterium obuense* was superior to *M. vaccae* and was much easier to manufacture with good manufacturing practice standards. Therefore, it was selected by Immodulon (IMM-101) as the immune modulator of choice for cancer studies.

IMM-101 is a systemic immune modulator containing a suspension of heat-killed whole cell *M. obuense*, a rapidly dividing, environmental and harmless saprophyte. The heat-killing treatment during manufacture safeguards patients from side effects associated with delivering live or attenuated organisms. IMM-101 has been successfully used in stage 4 melanoma trials as a single agent (Stebbing *et al.*, 2012). Furthermore, it has shown an ability to enhance responses to check point inhibitors (Dalgleish *et al.*, 2018) as well as to increase the effectiveness of Gemcitabine in a randomized study in patients with advanced pancreatic cancer where a significant survival and quality of life benefit was seen on the IMM-101/Gemcitabine arm *versus* Gemcitabine alone (Dalgleish *et al.*, 2016).

Of particular relevance to the development of Biovacc-19 is that the majority of patients who have received IMM-101 for advanced cancer have reported that, subsequently, they have not experienced the usual seasonal colds or influenzas from which they had previously suffered. IMM-101 has been administered to over 300 patients without a single serious side effect and is therefore a safe product to consider as a single priming adjuvant. Unpublished observations by these authors (Dalgleish) have suggested that this agent was the best adjuvant for producing effective immune responses against melanoma antigens.

A study to define an optimal antigen/adjuvant combination is in progress using a combination of Immodulon as a separate priming adjuvant together with a formulation of the vaccine peptides and an adjuvant. The study will define the balance between the adjuvant and antigen in such a way that one or a maximum of two doses will be needed to obtain protective immunity.

## The Biovacc-19 design concept and analysis of the target virus's general method of action for Infectivity

The Covid-19 vaccine Biovacc-19 is a peptide vaccine designed to develop antibodies to those parts of the SARS-CoV-2 spike protein which are engaged in binding and infecting cells.

The human SARS spike protein consists of two parts (Uniprot – P0DTC2, n.d.) https://www.uniprot.org/uniprot/P0DTC2. The S1 part attaches the virion to the cell membrane by interacting with host receptors like human ACE2 (Uniprot – Q9BYF1, n.d.) https://www.uniprot.org/uniprot/Q9BYF1 and with attachment receptors such as C-type Lectin domain family 4 member M (CLEC4M)/Dendritic Cell-Specific Intercellular adhesion molecule-3-Grabbing Non-integrin(DC-SIGNR) also known as CD209 (Marzi *et al.*, 2004; Uniprot – Q9H2X3, n.d.; Uniprot – Q9NNX6, n.d.; https://www.uniprot.org/uniprot/Q9H2X3 and https://www.uniprot.org/uniprot/Q9NNX6) thereby introducing the virus into the endosomes of the host cell, where the fusion peptide of the S2 part is unmasked and activated membrane

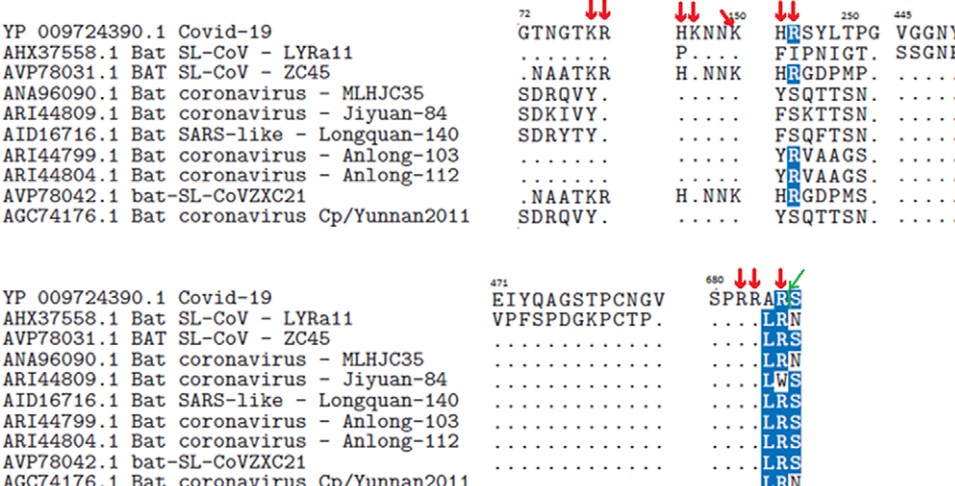

**Fig. 1.** Alignments of corona virus spike protein inserts.

fusion within endosomes occurs to permit virus replication in cytosol.

The SARS-CoV-2 spike is significantly different from any other SARS that we have studied (Lu *et al.*, 2020). The additional charge it carries [SARS-CoV-2 S1, isoelectric point (pI) pI = 8.24 *vs.* human SARS-CoV S1, pI = 5.67] will strongly improve the interactions with the receptor C-type lectin tail on CLEC4M/DC-SIGNR, which may, by itself, mediate the endocytosis of pathogens by acting as an attachment receptor, as happens for a number of other highly pathogenic viruses such as ebolavirus, Marburg, HIV-1, Hepatitis C, Measles, human CytoMegalo Virus, Influenza and others (Marzi *et al.*, 2004; Uniprot – Q9N2X3, n.d.; Uniprot – Q2NNX6, n.d.).

It is well documented that the receptor binding domain of the SARS-CoV-2 spike protein uses the ACE2 receptor. But clinical findings discussed below observed in Covid-19 patients suggest that other receptors for attachment such as CLEC4M/DC-SIGNR may be involved as well. We have investigated and sustained this supposition from amino acid-scale bio-chemical analysis.

Cumulative data suggests that the general method of action of this chimeric virus includes membrane components other than the ACE2 receptor, which may explain clinical evidence of its infectivity and pathogenicity. Data shows the nonspike receptor binding domain dependent phagocytic general method of action to be specifically related to cumulative charge from insertions on the SARS-CoV-2 spike (see Fig. 1) poised to form salt bridges with attachment receptors. This suggests that attachment to such previously reported membrane proteins has been enhanced directly due to the basic and positive charged inserts in the spike protein together with other basic and positive charged amino acid substitutions enabling formation of salt bridges with the receptor CLEC4M/DC-SIGNR or, indirectly, by the additional salt bridges formed between the positive charged amino acids and negative charged phospholipids on the cell membrane.

Positive-charged amino acids are inserted into peptides and proteins to enhance cell affinity and can also be used for transport of peptides and proteins through the cell wall (Thorén *et al.*, 2000; Richard *et al.*, 2003; Åmand *et al.*, 2011). In addition, these positive charges may be used for co-receptor binding where the opposite negative charge is available.

It is a matter of fact that there are unique inserts in the SARS-CoV-2 spike protein when they are aligned with other SARS-CoV sequences as shown in (Zhou *et al.*, 2020).

Fig. 1 shows six alignments with inserts. The first five inserts are pointed out by (Zhou *et al.*, 2020) and located near/around position 72, 150, 250, 445 and 471, respectively, whereas the insert around 680 is pointed out by (Coutard *et al.*, 2020) as a furin-like cleavage site with cleavage between R and S. Apart from inserts 4 and 5, these inserts are all basic inserts. The red arrows point out the basic amino acids. The green arrow and line point out the furin-like cleavage site.

It has been recently suggested that new X-Ray crystallography can assist in one of our key investigations: the docking of spike with receptor. (Shang *et al.*, 2020) Unfortunately, we cannot agree. The *Nature* paper entitled 'Structural basis of receptor recognition by SARS-CoV-2' does not, in fact, represent a true structure of the spike SARS-CoV-2 trimer apart from a modified part of the receptor binding motif (RBM). It uses the structure and the sequence for SARS-CoV deposited on 1 August 2005 as the backbone and then creates a chimera with the RBM (437–508) of SARS-CoV-2 modified and inserted. This is a confusing structural determination, representing neither virus. The authors were keen to focus on the importance of the ACE-2 receptor: 'we improved the ACE2-binding affinity of the chimeric RBD by keeping a short loop from the SARS-CoV RBM'. But then four of the six new charged inserts (1, 2, 3 and 6) outside the RBM were also excluded from their chimera, and the Cov-2 specific Cys538–Cys590 bridge which brings in additional charge from 526–560 (with pI = 10.03) via the Cys391–Cys525 right next to the RBM.

By these excisions the very essence of the novel structure and functionalities of SARS-Cov-2, and hence of its general mode of action for infectivity, is obscured. Therefore, using this structure from this paper would be quite misleading.

The SARS-CoV-2 3D structures have been usefully determined for SARS-CoV-2 trimer spike protein by (Walls *et al.*, 2020).

Our findings confirm (Coutard *et al.*, 2020) that the SARS-CoV-2 contains a furin-like cleavage site absent in CoV of the same clade. Also in Fig. 1, Coutard highlight that enriched basic charge associated with this cleavage site are found in a number of viruses such as HIV, influenza, human CytoMegalo virus (Herpes) and respiratory syncytial virus, yellow fever, zika and ebola. Coutard *et al.* (2020) furthermore state that, 'conversely, the highly pathogenic forms of influenza have a furin-like cleavage site cleaved by different cellular proteases, including furin, which are expressed in a wide variety of cell types allowing a widening of the cell tropism of the virus'. Furthermore, the insertion of a multibasic motif RERRRKKR↓GL

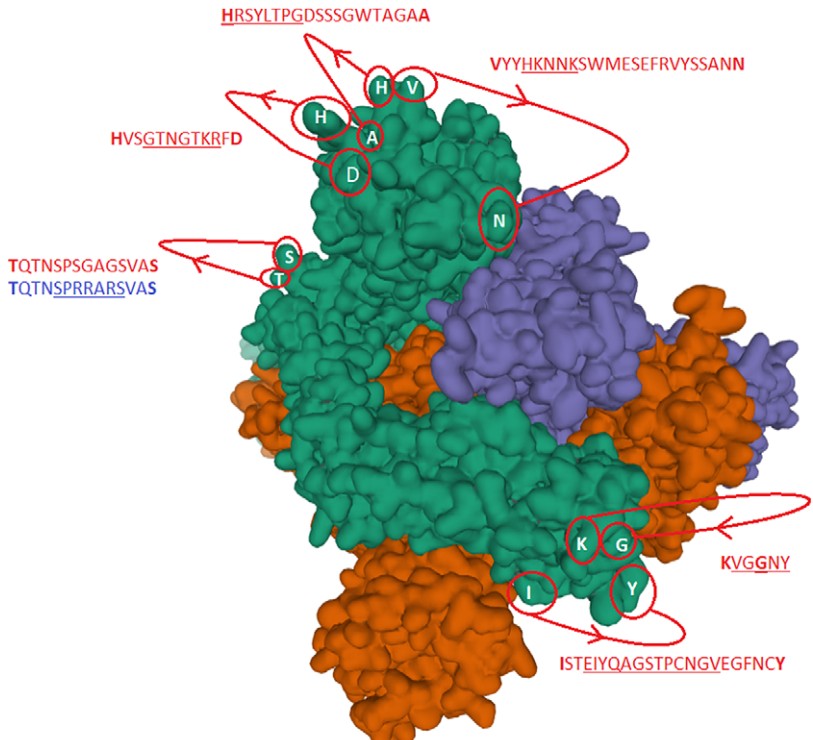

**Fig. 2.** The identified inserts examined in the PDB 6VXX electron microscopy structure (Walls *et al.*, 2020) The sequences highlighted in red could not be found in the cryo-electron microscopy structure data. The six aligned sequences in Fig. 1 are underlined in the missing sequences. Bold amino acids indicate first and last amino acids used to build the structure where the missing part is in between. Insert 6 did not have the same sequence in 6VXX as in the reference Sars-CoV-2 sequence. The authors stated that a designed mutated strain lacking the furin cleavage site residues was used.

at the H5N1 hemagglutinin HA cleavage site was likely associated with the hyper-virulence of the virus during the Hong Kong 1997 outbreak. Extensive clinical evidence in this pandemic suggests that SARS-CoV-2 poses such widened cell tropism.

The mechanism of action linked to such basic Arginine rich domains is known as the binding of cell-penetrating peptides (Thorén *et al.*, 2000). The important point to grasp is that such positively charged amino acids need to be located in such a way that they span four amino acids (or more) in length to act as an initial membrane anchor. Use of such positive charged vaccine peptides allows for attachment and/or direct cell uptake depending on the net charge present in the peptide (Åmand *et al.*, 2011; US patent – US9950811B2n). The present authors have used such basic properties in uploading vaccine peptides to cells (typically macrophages and dendritic cells). We have found that more than three Arginines are required for uptake. In addition, this charge needs to be distributed over the peptide. (Yesylevskyy *et al.*, 2009). Fig. 3a–f illustrate the process in dynamic sequence from attachment to cell membrane until complete cell penetration.

For SARS-CoV-2, the sequence (SPRRAR|S) is longer and more basic than the SARS-CoV (TVSLLR|S) and hence is more potent (Coutard *et al.*, 2020). The mode of action of a furin-like cleavage site is, following endosomal encapsulation, to facilitate attachment and penetration of the inside wall of the endosome to release the uncoated virus into cytosol where it can start replication and release its full pathogenic potential. In the context of SARS-CoV-2 having a general elevated pI with additional charge located in the receptor binding domain as shown below, will make it fit for membrane penetration.

The co-receptor dependent phagocytic general method of action of SARS-CoV-2 appears to be specifically related to cumulative charge: please refer to SARS-CoV-2 peaks above pI = 8.24 (Fig.5)

compared to human SARS-CoV (Fig. 4). These basic domains – partly inserted and partly substituted amino acids – explain the salt bridges formed between the SARS-CoV-2 spike and its co-receptors on the cell membrane. Indeed, these data suggest that the infectivity of SARS-CoV-2 is best explained by this cumulative charge associated with these basic charged domains, enabling extra salt bridges to attach to membrane components as well as to the membrane itself.

Cys131–Cys166 loop 1 is partially missing in the electron microscopy structure. The missing section is highlighted in blue in Fig. 7. No information is given to explain why the SARS-CoV-2 sequence was changed for the electron microscopy. When preparing samples for examination, perhaps it may have been necessary to alter the surface of trimer containing a surplus of hydrophilic and basic/positive amino acids by removing sequences as highlighted in Figs 2 and 5. By making visible the complete repertoire in this paper, we are therefore able to display data that, taken together, present the general mechanism of action necessary for understanding how the SARS-CoV-2 virus enter cells and hence where and how to attack it with a vaccine. With absence of complete repertoire, it is significantly more difficult to understand the general method of action and therefore to find a vaccine or a therapy.

Underlined sequences are identified charged amino acids contributing to attachment/co-receptor binding such as CLEC4M/DC-SIGNR(CD209). The exposed and positive charged amino acids (in red capital letters) contributing to salt bridges are located within the Cys131–Cys166 loop 1: (KV**C**EFQFCNDPFLGVYY<u>HKNNK</u>S WMESEFRVYSSANN**C**TF) and Cys336–Cys361 loop 2: (NL**C**PFGE VFNAT<u>RFASVYAWNRKRIS</u>N**C**VA). This second charged domain (Fig. 7a) is positioned right next to the receptor binding motif (437–508) via the adjacent Cys379–Cys432 bridge and

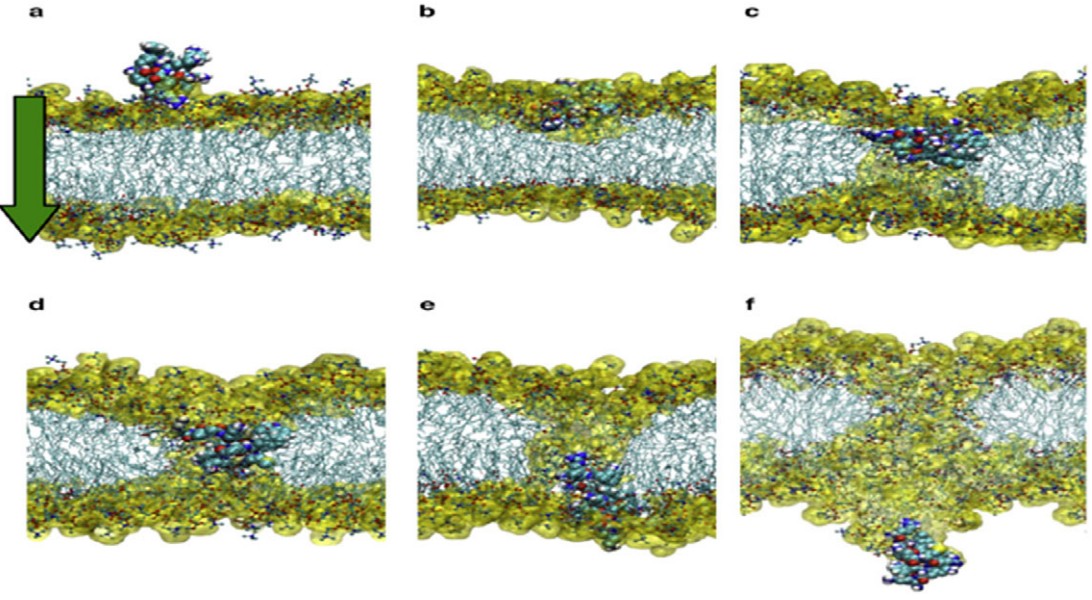

**Fig. 3.** From (Yesylevskyy *et al.*, 2009).

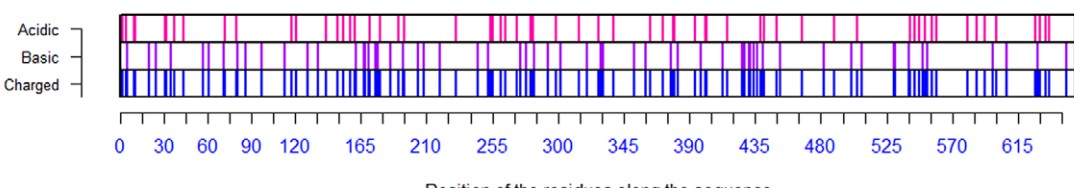

Position of the residues along the sequence

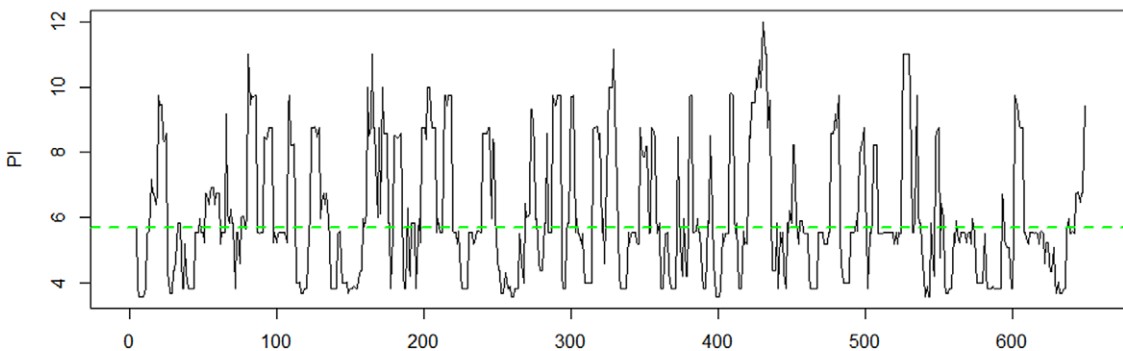

**Fig. 4.** Distribution of charge using a rolling window of 12 amino acids in steps of 1 on human SARS-CoV shows a dominance of acidic amino acids giving a pI = 5.67 (green dotted line). Repulsion will be observed in the presence of a protein/peptide domain with a similarly low pI = 5.12 as found for ELEC4M/DC-SIGN. Ref.: https://www.uniprot.org/uniprot/P59594 sp|P59594|14–667 [SARS-CoV Urbani].

can therefore facilitate binding to opposite charged attachment receptors. The domain 526–560 (pI = 10.03) is brought into the receptor binding domain via Cys391–Cys525 bridges furthermore significantly enhances the overall charge on the receptor binding domain (**C**GPKKSTNLVKNK**C**VNFNFNGLTGTGVLTESNKKFL). This additional Lysine (K) driven charge on SARS-CoV-2 coming from the domain 526–560 does not exist on SARS-CoV due to the unique SARS-CoV-2 Cys538-Cys590 bridge. The RBD Cys131–Cys166 loop1 on SARS-CoV-2 has a pI = 5.34 while the similar Cys128–Cys159 loop2 on SARS-CoV has a pI = 4.36. For the SARS-CoV-2 Cys336–Cys361 domain, we have found a pI of 9.5, whereas for the

similar SARS-CoV 323–348 we have a pI of 8.82. As shown here, the surface exposed cysteine loops on the RBD have consistently higher pI for SARS-CoV-2 than for SARS-CoV.

This method of action using ACE2 as main receptor and ELEC4M/DC-SIGN as co-receptors is similar to what is observed for HIV and its use of CD4 as main receptor and the V3 Cys–Cys loop docking on the CCR5/CXCR4 co-receptors. A substitution of the amino acid Cys348 with an Ala for SARS-CoV leads to complete loss of human ACE2 binding *in vitro* (Wong *et al.*,2004; Uniprot – P59594, n.d.). The construction of these Cys–Cys bridges in SARS-CoV-2 are similar. So, would it be reasonable to assume that a

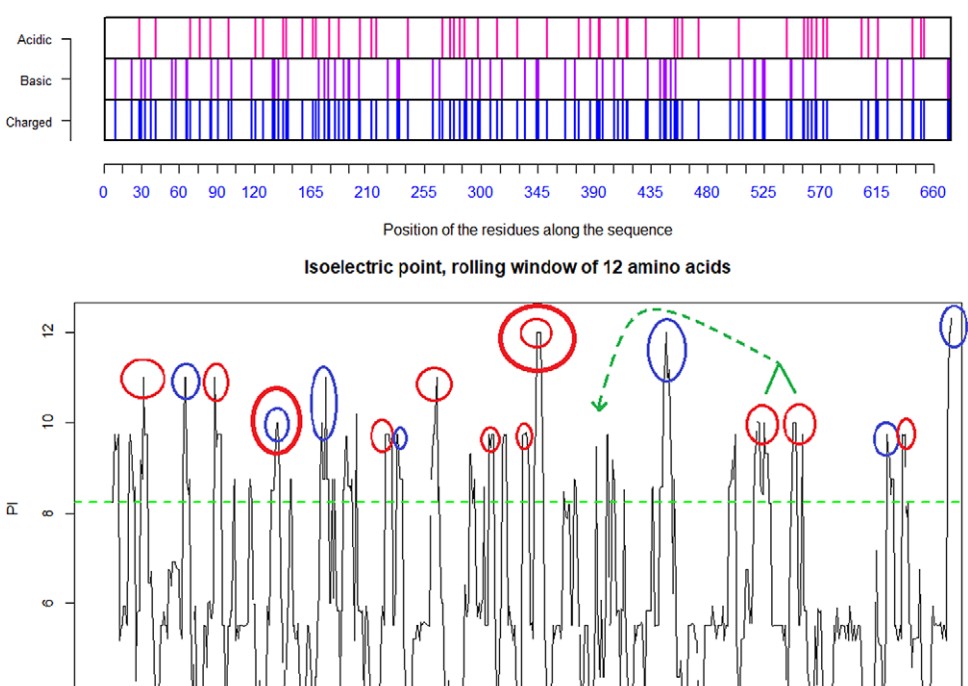

**Fig. 5.** Distribution of charge using a rolling window of 12 amino acids in steps of 1 on SARS-CoV-2 shows a dominance of basic amino acids. The red circles are verified to be surface exposed while the blue circles are missing in PDB: 6VXX. Non-labelled peaks are not surface exposed. Above the green dotted line (Isoelectric point for the SARS-CoV-2 spike protein S1 pI = 8.24) the peaks contain basic residues which can form salt bridges in the presence of acidic amino acids such as in the receptor CLEC4M. The Cys131–Cys166 and Cys336–Cys361 loops are highlighted by as bold red/blue and red/red circles. Furthermore, the Cys391–Cys525 bridge: a highly basic domain (526–560) with pI = 10.03 is moved forward and sits next to the receptor binding motif (408–508) as part of a Cys538–Cys590 loop, as indicated by the green arrow. Ref.: https://www.uniprot.org/uniprot/P0DTC2 sp|P0DTC2| 13–685 [SARS-CoV-2].

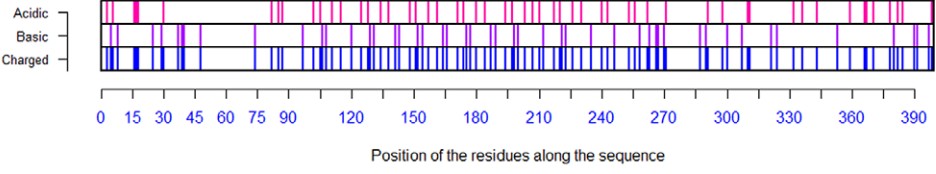

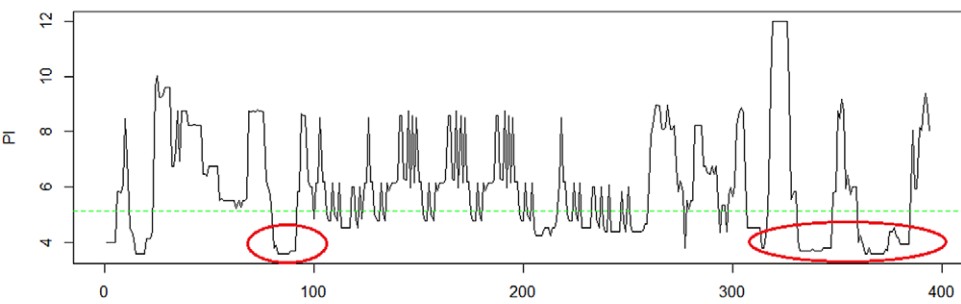

**Fig. 6.** Distribution of charge using a rolling window of 12 amino acids in steps of 1 on the receptor CLEC4M having pI = 5.12. The ovals identify the most likely contributors to the salt bridges between the acidic domain, here on CLEC4M and the highlighted basic domains on the SARS-CoV-2 S1 protein in Fig. 3 above. The two disulfide bonds Cys296–Cys389 and Cys368–Cys381 present in the C-type lectin C-terminal part of CLEC4M pull back the Cys368–Cys381 acidic domain to position 296 and make that domain a highly condensed with acidic amino acids ready for formation of salt-bridges with the basic amino acids in S1 of SARS-CoV-2 (Marzi *et al.*, 2004). Further investigations might possibly show that other amino acids on the S1 SARS-CoV-2 are involved in such attachment receptor binding. Ref.: CLEC4m/DC-SIGN – also referred to as CD209 https://www.uniprot.org/uniprot/Q9H2X3.

similar substitution of Cys361 with an Ala for SARS-CoV-2 would lead to a similar loss of human ACE2 binding? The answer is probably yes for ACE2 binding. But due to the additional cumulative charge it would still be able to attach to the co-receptor.

This is an important finding for our vaccine design. Furthermore, that SARS-CoV-2 can enter cells without using the ACE2, but also by promiscuous attachment has implications for understanding disease epidemiology, for treatment drug method of

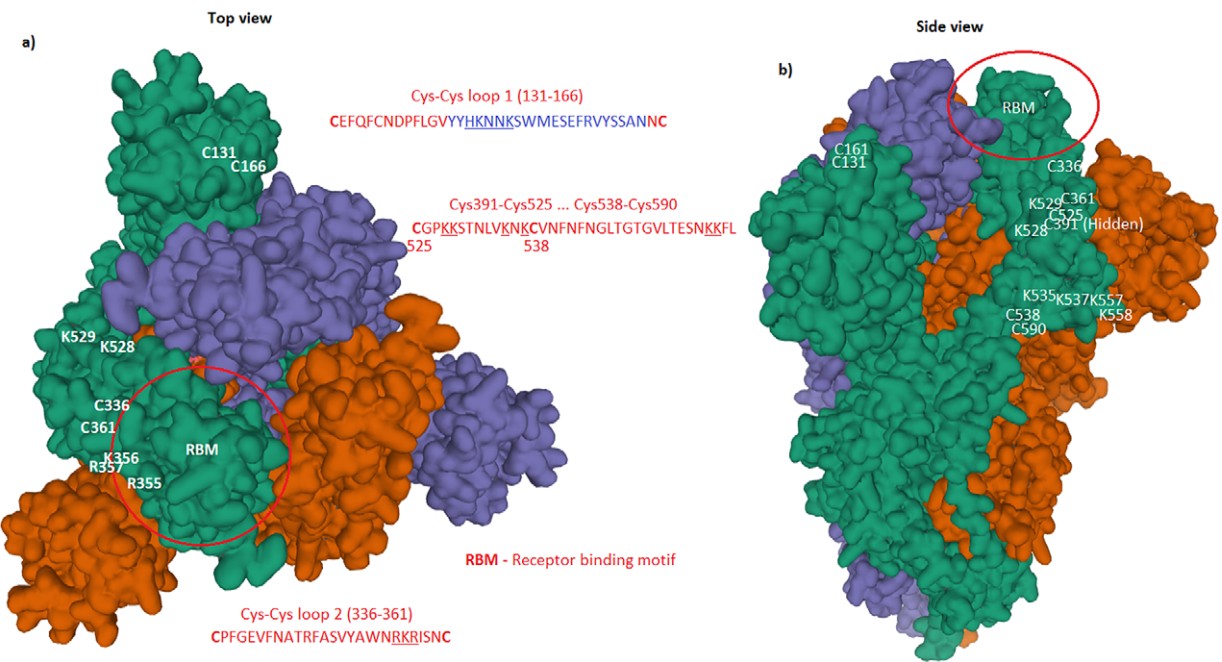

**Fig. 7.** Spike trimer (a) top view and (b) side view. The specific receptor binding motif (RBM) is located on the sequence (437–508), while the receptor binding domain (RBD) has a broader location (319–541); Ref. https://www.uniprot.org/uniprot/P0DTC2. The charged cysteine associated domains are Cys131–Cys166, Cys336–Cys361, Cys391–Cys525 … Cys538–Cys590. As can be seen, there is a high concentration of positive charged surface exposed amino acids within the receptor domain next to the RBM.

action as well as for vaccine development strategies. Similar co-receptor observations have been made before. Wong *et al.* (2004) report, interestingly, the first 330 amino acids of the 769-residue S1 subunit of the mouse hepatitis virus (MHV) S protein is sufficient to bind carcinoembryonic antigen-related cell adhesion molecule 1 (CEACAM1), to the cellular receptor for the mouse hepatitis virus (MHV). Furthermore, a different region of the S1 domain of HCoV-229E, between residues 407 and 547, is sufficient to associate with the cellular receptor for this coronavirus, aminopeptidase N (APN, CD13).

In particular, the insert in alignment 6 in the SARS-CoV-2 spike has three positive Arginines in combination with a Proline, which together secure the anchoring to the membrane (but not acting in the same way as a typical cell penetration peptide due to there being only four amino acids). Therefore, these data show that the molecular structure of the SARS-CoV-2 spike receptor binding domain, with its accumulated charge from inserts and salt bridges in surface positions, is capable of binding with cell membrane components. This is an essential key to understanding its potency.

Furthermore, the Covid-19 pandemic is revealing neurological, haematological and immunological pathogenicity in the virus which cannot be explained by the ACE2 receptor alone. Profuse clinical observations of loss of taste, smell, sore throat, dry cough and headache and severe stomach/gastrointestinal pain with diarrhoea arising in the pandemic are evidence that early phase Covid-19 is binding to the bitter/sweet receptors which also provides a perfect location for follow on transmission by coughing. How is it doing this? The answer to these clues is also important for our vaccine design because, as explained in the next section, it caused us to select and deploy epitopes accordingly in order to deny the virus these binding options.

Compromising the function of olfaction and bitter/sweet receptors will effect a timely release of products from the innate immune system, thereby enhancing infectivity and transmission. At the onset of infection, there is enhanced mucociliary transport of mucus from the nasopharynx or oropharynx. By swallowing innate immune products at the same time, they are disseminated on the airway surface (Workman *et al.,* 2015). These immune products include directly anti-microbial compounds such as defensins, lactoferrin, cathelicidins and lysozyme, in addition to reactive oxygen species and nitric oxide that also display potent antimicrobial activity (Roper *et al.,* 2013). Several indirect pathways are activated as well, with the release of cytokines and chemokines that recruit the adaptive immune system and begin inflammatory cascades.

SARS-CoV-2 may use a direct route too. Attached to epithelial cells in the oral cavity, potentially it may also be transported together with food via the stomach to the intestine. As noted, clinicians report severe stomach/gastrointestinal pain with acute diarrhoea in Covid-19 patients. Such clinical indicators support our earlier analysis and also suggest that SARS-CoV-2 can use other attachment/co-receptors than ACE2.

Further collateral support for these hypotheses came in 2018 when Zhou *et al.* (2018) isolated a new coronavirus which they named SADS (Swine Acute Diarrhoea Syndrome). They investigated receptor usage in the intestines of infected piglets but could find no evidence of involvement of any of the three receptors known from previous SARS epidemics: ACE2, APN and dipeptidyl peptidase 4. As with SADS, the similar pathology associated with Covid-19 points in the direction of a different and more promiscuous attachment/co-receptor like CLEC4M/DC-SIGN driven by a positively charged spike trimer surface.

We have earlier explained the enhanced presence of basic amino acids in the inserts such as Lysine (K) and Arginine (R) and their association with enhanced pathogenicity in other pathogens like the 1997 H7N1 Hong Kong Flu (Kido *et al.,* 2012; Coutard *et al.,* 2020). We noted above the critical importance of understanding that cumulative positive charge associated with the inserted short sections has the effect of enabling extra salt bridges to attach to the

membrane. Under this general method of action, this combination of basic amino acids in the SARS-CoV-2 spike binds to cells in the upper airways. Its high infectivity is associated with olfaction and taste; and systemic release of the virus explains the clinical findings associated with destruction of erythrocytes (Liu *et al.*, 2020), T-cells and cells associated with neuropathological conditions (Henry *et al.*, 2020).

From all these convergent data, we therefore posit that the general method of action for SARS-CoV-2 is indeed as a co-receptor dependent phagocytic process: the cell envelops the virus due to its opposite charged binding to co-receptors on the cell membrane such as CLEC4M/DC-SIGNR and possibly to the membrane itself. But, furthermore, simultaneously, it is capable of binding to ACE2 receptors in its receptor binding domain: in fact, SARS-CoV-2 is possessed of dual action capability.

Once the virus is phagocytosed, it will take over cell machinery, replicate and kill the host cells and rapidly increase systemic infection. The virus is thereby killing off erythrocytes which would account for the hypoxia observed in advanced patients, which leads to the shortage of oxygen uptake which may eventually prove fatal. Furthermore, clinically identified involvement of this class of olfaction and bitter/sweet receptors as potential co-receptor(s), or as alternative receptor sites to ACE2, has possible implications for binding, the replication, metabolism and pathology of the SARS-CoV-2 virus.

Although undertaken for the purposes of vaccine design, these virological findings have major positive implications for enhanced treatment options for advanced or relapsed Covid-19 patients, five of which such treatments one of us in his clinician role (Dalgleish) has published elsewhere (Dalgleish *et al.*, 2020).

## Specific implications from the target virus's general method of action for design of Biovacc-19

In order to ensure that Biovacc-19 covers all the various cell receptor binding options, combined with our guiding criterion of using NHL epitopes, we systematically blasted (Uniprot – P0DTC2, n.d.) the spike protein. These NHL epitopes are displayed in Table 1.

**Table 1.** Nonhuman-like (NHL) sequences found in SARS-CoV-2 spike protein S1.

| Seq no. | Sequence | Seq no. | Sequence | Seq no. | Sequence | Seq no. | Sequence |
|---|---|---|---|---|---|---|---|
| 1 | PLVSSQ | 41 | SKHTPI | 81 | GKIADY | 121 | GVLTES |
| 2 | SSQCVN | 42 | LVRDLP | 82 | KIADYN | 122 | TESNKK |
| 3 | VNLTTR | 43 | RDLPQG | 83 | DDFTGC | 123 | PFQQFG |
| 4 | LTTRTQ | 44 | PQGFSA | 84 | FTGCVI | 124 | QQFGRD |
| 5 | TRTQLP | 45 | GFSALE | 85 | GCVIAW | 125 | ADTTDA |
| 6 | PAYTNS | 46 | VDLPIG | 86 | LDSKVG | 126 | DTTDAV |
| 7 | VFRSSV | 47 | PIGINI | 87 | DSKVGG | 127 | TTDAVR |
| 8 | VLHSTQ | 48 | GINITR | 88 | KVGGNY | 128 | DAVRDP |
| 9 | LFLPFF | 49 | LTPGDS | 89 | GGNYNY | 129 | AVRDPQ |
| 10 | SNVTWF | 50 | TPGDSS | 90 | KSNLKP | 130 | DITPCS |
| 11 | VSGTNG | 51 | AAYYVG | 91 | PFERDI | 131 | ITPCSF |
| 12 | SGTNGT | 52 | GYLQPR | 92 | ISTEIY | 132 | TPCSFG |
| 13 | TNGTKR | 53 | ALDPLS | 93 | STEIYQ | 133 | FGGVSV |
| 14 | NGTKRF | 54 | LDPLSE | 94 | EIYQAG | 134 | GGVSVI |
| 15 | RFDNPV | 55 | PLSETK | 95 | STPCNG | 135 | SVITPG |
| 16 | VYFAST | 56 | SETKCT | 96 | TPCNGV | 136 | ITPGTN |
| 17 | ASTEKS | 57 | KCTLKS | 97 | PCNGVE | 137 | TPGTNT |
| 18 | STEKSN | 58 | TVEKGI | 98 | GVEGFN | 138 | PGTNTS |
| 19 | IRGWIF | 59 | TSNFRV | 99 | PLQSYG | 139 | TSNQVA |
| 20 | WIFGTT | 60 | FRVQPT | 100 | FQPTNG | 140 | VAVLYQ |
| 21 | FGTTLD | 61 | TESIVR | 101 | TNGVGY | 141 | QLTPTW |
| 22 | TTLDSK | 62 | SIVRFP | 102 | GVGYQP | 142 | STGSNV |
| 23 | LDSKTQ | 63 | PNITNL | 103 | LLHAPA | 143 | GSNVFQ |
| 24 | DSKTQS | 64 | ITNLCP | 104 | LHAPAT | 144 | FQTRAG |
| 25 | SKTQSL | 65 | NLCPFG | 105 | HAPATV | 145 | QTRAGC |
| 26 | KTQSLL | 66 | LCPFGE | 106 | APATVC | 146 | RAGCLI |
| 27 | VNNATN | 67 | ATRFAS | 107 | PATVCG | 147 | AGCLIG |
| 28 | ATNVVI | 68 | TRFASV | 108 | ATVCGP | 148 | AEHVNN |

**Table 1** Continued

| Seq no. | Sequence | Seq no. | Sequence | Seq no. | Sequence | Seq no. | Sequence |
|---|---|---|---|---|---|---|---|
| 29 | CEFQFC | 69 | SNCVAD | 109 | TVCGPK | 149 | IPIGAG |
| 30 | FCNDPF | 70 | VLYNSA | 110 | VCGPKK | 150 | AGICAS |
| 31 | CNDPFL | 71 | FKCYGV | 111 | GPKKST | 151 | SYQTQT |
| 32 | LGVYYH | 72 | KCYGVS | 112 | PKKSTN | 152 | QTQTNS |
| 33 | GVYYHK | 73 | CYGVSP | 113 | KKSTNL | 153 | TQTNSP |
| 34 | VYSSAN | 74 | YGVSPT | 114 | KSTNLV | 154 | TNSPRR |
| 35 | SANNCT | 75 | GVSPTK | 115 | VKNKCV | | |
| 36 | YVSQPF | 76 | VSPTKL | 116 | FNFNGL | | |
| 37 | VSQPFL | 77 | ADSFVI | 117 | FNGLTG | | |
| 38 | LEGKQG | 78 | QIAPGQ | 118 | NGLTGT | | |
| 39 | EGKQGN | 79 | APGQTG | 119 | TGTGVL | | |
| 40 | GKQGNF | 80 | TGKIAD | 120 | GTGVLT | | |

These sequences were obtained by blasting the spike protein sequence using moving window of 6 amino acids in steps of 1 against the human protein sequence database on Uniprot (Uniprot – P0DTC2, n.d.).

**Table 2.** Probability of success of different vaccine technologies.

| Property\vaccine technology | Synthetic peptides | Vector based (virus or RNA) |
|---|---|---|
| Epitope targeting receptor binding domains (neutralization) | Specifically selected | No selection. The immune system of each individual will select and present the most dominating epitopes |
| Epitope presented and likelihood for getting local or systemic toxicity (SAE) | Low to very low. In theory no SAE should be observed since all epitopes are non-human like. | Difficult to predict The epitopes presented will be a mixture of human like (78.4%) and non-human like (21.6%) epitopes |
| Antibody-dependent enhancement (ADE) | Low Since the antibodies are directed towards the receptor binding domain and other co-receptor domains | Difficult to predict In such vaccine designs, there is no innate guiding of where the antibodies should bind. However, due to continued boosting of these epitopes through life, there is an elevated risk for development of ADE which must be expected due to the fact that if the virus returns at a later date in a mutated form, having modified antigenic composition, partial binding may occur and hence result in ADE (Ricke D, et al., 2020; Negro et al., 2020). |

Peptide vaccine antigens used in Biovacc-19 are peptide strings with a total length of 30–36 amino acids consisting of epitopes placed in scaffolds similar to those described in US patent US9950811B2n (US patent – US9950811B2n). The precise sequences are not disclosed here.

Generation of antibodies requires TH-2 responses. By using a set of peptides spanning more than 130 amino acids it will be as if a medium sized protein was used as vaccine antigen resulting in large sequences variation and hence giving a surplus of TH-2 epitopes.

The design of such synthetic vaccine peptides offers a further advantage. It will permit discrimination between antibody responses coming from natural infection and those coming as a result of vaccination.

A group of four of these vaccine peptides have been tested for acute toxicity with success.

The Biovacc-19 vaccine is based on the method of action described. The vaccine peptides have therefore been selected from such NHL epitopes located in or close to the charged inserts and to the expected co-receptor binding locations outside the main receptor binding domain in addition to the NHL epitopes available for use within the receptor binding domain.

The benefit of using this strategy compared to conventional virus, ribonucleic acid (RNA) or other vector-based vaccine systems is that the immune system will be guided directly to the epitopes which are relevant for virus neutralization. A further advantage of using NHL epitopes is that the immune system is free to mount robust, broad and long-lasting immune responses without being limited by local or even systemic immune-toxic reactions against our own human protein epitopes.

A comparison of the three most relevant properties/factors used for determining the probability of success in a vaccine design is presented in Table 2.

The patented vaccine peptides constituting Biovacc-19 are a combination of various sequences of five amino acids found on the spike protein. The scaffold design of the vaccine peptides will create 'in-between' epitopes which will facilitate discrimination between vaccine induced antibodies and antibodies which result from exposure to a Covid-19 infection.

Since successful acute toxicity tests have already been performed and since safe NHL epitopes are used, it is logical for the synthesis of Active Pharmaceutical Ingredients under Good Manufacturing Practice while optimizing the combination of antigen doses and adjuvants and preparing for manufacture at scale to be undertaken in parallel.

It is envisaged that Biovacc-19 will be ready for human clinical trials in the fourth quarter of 2020 although acceleration will possible, given strategic funding to do so.

## Conclusion

We have offered a rationale for the design methodology and the necessary design parameters of a successful and safe vaccine against SARS-CoV-2. It is not included in any of the eight vaccine design routes identified in a recent *Nature* summary graphic (Callaway, 2020). We have shown in this paper why a comprehensive analysis of the aetiology of the target virus is prerequisite, not optional. From the HIV experience, we have illustrated the risks of not doing so.

Next, we explained why, unlike in conventional vaccine design procedures, the choice of adjuvant is not to be seen as an after-thought but as integral from the beginning. We have deliberately chosen an adjuvant which has been shown to activate the innate and cell-mediated immune responses which are crucial to the successful presentation of the relevant epitopes. We have shown how Biovacc-19 has employed our understanding of the general method of action for infectivity and pathogenicity of the target virus to optimize action and to minimize risk, especially ADE; and we have presented the NHL epitopes in the SARS-CoV-2 spike from which Biovacc-19 has been down-selected.

**Open Peer Review.** To view the open peer review materials for this article, please visit https://doi.org/10.1017/qrd.2020.8.

**Acknowledgements.** The authors acknowledge invaluable assistance with analysis and critical comment by Dr. J.-F. Moxnes of the Norwegian Defence Research Establishment, Kjeller, Norway.

**Authorship contributions.** Writing original draft: B. S.; Review and editing: A. D.; Investigation and review and editing: A. S.

**Financial support.** This research received no specific grant from any funding agency, commercial or not-for-profit sectors.

**Conflict of interest.** Birger Sørensen is an employee and shareholder of Immunor AS. Andres Susrud is an employee and shareholder of Immunor AS. Angus Dalgleish is on the scientific advisory board of Immodulon, and has stock options in Immunor AS.

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
