## [Reviewer Report]

*Comments to Editor*: Sørensen et al report a very interesting and timely bioinformatic analysis and study “Method of Action of Biovacc-19” in pre-clinical trial development. Biovacc-19 is a candidate vaccine for Covid-19 (SARS-COV-2) targeting the spike S1 protein that recognizes the ACE2 receptor for entry into cells for its mode of infection. Their strategy is to design the most effective vaccine that is both safe and able to combat SARS-COV-2. They focus on finding nonhuman-like (NHL) sequences 21.6% in the SARS-COV-2 spike S1 protein, rather than the 78.4% sequences already existing in humans. This strategy is sound and their approach may not only stimulate a good immune response, but also avoid triggering the human’s own immune system into attacking its own proteins, resulting in autoimmune diseases.

Their analysis is sound and thorough. Through detailed alignment and attention to detail, they have uncovered some features that previous analyses missed.

There are some minor points to improve the manuscript. After they make the changes and revise the manuscript, this reviewer highly recommends expedient publication in QRB Discovery.

*Comments to Author*: Sørensen et al report a very interesting and timely bioinformatic analysis and study “Method of Action of Biovacc-19” in pre-clinical trial development. Biovacc-19 is a candidate vaccine for Covid-19 (SARS-COV-2) targeting the spike S1 protein that recognizes the ACE2 receptor for entry into cells for its mode of infection.Their strategy is to design the most effective vaccine that is both safe and able to combat SARS-COV-2. They focus on finding nonhuman-like (NHL) sequences 21.6% in the SARS-COV-2 spike S1 protein, rather than the 78.4% sequences already existing in humans. This strategy is sound and their approach may not only stimulate a good immune response, but also avoid triggering the human’s own immune system into attacking its own proteins, resulting in autoimmune diseases.

Sørensen et al base on their strategy on the analysis of others’ experience of developing HIV vaccine.They point out that despite years of effort, many-trial-and error studies, and a large amount of investment and clinical trials, an effective HIV vaccine still remains elusive.

To avoid taking the same path, they focus on nonhuman-like sequences. If their strategy works well for SARS-COV-2, their method could be generalized for many vaccines, both viral and nonviral pathogens.

Their analysis is sound and thorough. Through detailed alignment and attention to detail, they have uncovered some features that previous analyses missed. They noticed the pI differences between SARS-COV-1 and SARS-COV-2 and other similar coronaviruses. They found that the positively-charged amino acids lysine (K) and arginine (R) are in the specific locations that can form ionic bonds (salt bridges) to stabilize the S1 protein. They specifically pointed out the additional 3K and 3R may play a key role as an effective antigen to elicit body’s immune response.

They also emphasize that one crucial aspect for a successful vaccine is the choice of adjuvant which must be very carefully chosen as integral to the vaccine strategy from the beginning, not as an afterthought. Few people have paid such detailed attention explicitly. This reviewer believes they are correct in this regard from carefully reading the literature.

They provide a detailed Table 1 of nonhuman-like sequences found in SARS-COV-2. They systematically analyze the protein sequence of the SARS-COV-2 in 6-amino acid segments reviewing a total 155 segments. These segments are important for vaccine development as they demonstrated for Biovacc-19. They also compared various methods of generating vaccines and ranked the likelihood of success in Table 2, to provide some guidance for vaccine development.

There are some minor points to improve the manuscript:

1) They use many acronyms without defining them first and the reader quickly gets lost. This reviewer needed to check the literature and Wikipedia to understand these acronyms. For example: FcgR (p.2), BCG (p. 3), IMM-101 (p.3), CLEC4M/DC-SIGNR (p.4), CMV (p.4). It is common in immunology to use many acronyms.But the non-experts are quickly confused and have to check the literature to understand these acronyms before understanding their meaning. The late legendary Francis Crick advised this reviewer in 1991 “If you do not have to use acronyms, especially only two or three words, it is best to use the words, rather than acronyms”. This reviewer took this advice to heart.

2) The authors should provide a more detailed legend for Figure 1. They should also use arrows to pinpoint the key 3 K and 3 R in the SARS-COV-2 sequence for clarity. The authors should be explicit and show these amino acids in the well-refined X-ray crystal structure. Please see: https://pubmed.ncbi.nlm.nih.gov/32225175/

3) The authors should explain in the Table 1 legend how these peptide sequences are obtained; whether shifting two to five amino acids from the previous peptide sequence, or otherwise.It is best not to make the reader figure this out.The authors should be explicit.

4) Similarly, they should provide more detail for the legends of Figure 4, 5 and Figure 7.

5) There is a new paper Structural basis of receptor recognition by SARS-CoV-2of a 2.68Å resolution of crystal structure just published in Nature, March 2020. It has more details than the CryoEM structure and show the exact location of interface between spike S1 and ACE2 receptor including several arginines.This reviewer suggests the authors use this structure to pinpoint where the 3 K and 3 R and other nonhuman-like sequences are located in the structure.

6) Please cite this reference:

Jian Shang et al Structural Basis of Receptor Recognition by SARS-CoV-2. 2020 May; 581(7807): 221-224. doi: 10.1038/s41586-020-2179-y. Epub 2020 Mar 30, 2020.

https://pubmed.ncbi.nlm.nih.gov/32225175/

---

## [Reviewer Report]

*Comments to Author*: The need for a vaccine is urgent as the Covid-19 pandemic continues to spread. “Biovacc-19”, from the company Immunor, is a peptide vaccine candidate targeting selected non-human-like epitopes of the SARS-CoV-2 spike protein. The manuscript “Biovacc-19: A Candidate Vaccine for Covid-19 (SARS-CoV-2) Developed from Analysis of its General Method of Action for Infectivity” by Sørensen, Susrud, and Dalgleish is based on the central idea that additional cationic amino acids on the SARS-Cov-2 spike protein (compared to other coronaviruses) increases its infectivity through electrostatic attraction to biomembranes and non-ACE2 cellular targets. Albeit an educated guess, in the absence of verification experiments, such a mechanism of action is still only a guess. Sørensen et al. should clearly state that the proposed mechanism of action is only based on observing the protein sequence and mapping charged amino acids on a 3D model of the spike protein.

Nevertheless, there are no peptide vaccines among the competitors currently in advanced clinical trials. Because Biovacc-19 represents an innovative and very different vaccine development strategy, I recommend the manuscript from Sorensen et al. to be published in QRB Discovery with minor revisions. Especially the selection for non-human-epitopes is a major laudable step in the development process of new vaccines. The manuscript is written less like a scientific report, and more as a justification for a product from Immunor already undergoing initial tests. However, the format of QRB Discovery should be able to accomodate such manuscripts.

The reviewer randomly found out that the section “Coutard et al furthermore state that conversely, the highly pathogenic … Hong Kong 1997 outbreak.” (after the word ‘that’) is verbatim from Coutard et al., but the length of the quote and the lack of quotation marks increases the risk of confusion. All verbatim citations in the manuscript should be clearly marked as such.

It is somewhat unclear if the authors imagine that the arginine rich insert 6 “RRAR” is membrane penetrating, or only acts as a membrane anchor on the surface. Figure 3 shows the action of a cell penetrating peptide, but they also write later “but not acting in the same way as a typical cell-penetration-peptide”. Does “RRAR” stay on the surface of the membrane, or cross the membrane boundary, and how does the activity change upon cleavage at the adjacent furin site?

From a physical perspective, the total number of positively charged amino acids in the six inserts (compared to SARS) is still very limited: KR in insert 1, HKK in insert 2, HR in insert 3, none in insert 4, none in insert 5, and RRR in insert 6. The manuscript mentions the unspecific effect of 7 positive charges from inserts 1-5 spread all over the protein (the basis of the higher isoelectric point of the spike protein of SARS-CoV-2 compared to SARS-CoV), and also the specific effect from the three arginines concentrated in insert 6. The two are fundamentally different. Comparing the two, should the latter be more relevant and dominating? If insert 6 really acts as a cell penetrating peptide in some way, why is “RRAR” only reported as a furin cleavage site in the literature, when occurring in other viruses?

Key references are missing mainly to Thorén and Åmand who were first to explain the basic uptake mechanisms of peptides, including the electrostatic effects that the authors claim they exploit. Some references directly related to the current study: https://doi.org/10.1016/S0014-5793(00)02072-X and https://doi.org/10.1016/j.bbamem.2011.03.011

The two charged domains, in the two Cys-Cys loops, should be marked out in Fig 5 for clarity.

There is very little proof in the literature that “it is evident that early phase Covid-19 is binding to the bitter/sweet receptors which provide a perfect mechanism for spread”. The virus could alternatively attack nerve cells, and indrectly destroy the taste buds, without directly binding to the receptors. Since this is a mechanism on the molecular level, the authors should expand on this claim and add more justification.

Likewise, there is very little proof that “The virus is thereby killing off erythrocytes which would account for the hypoxia observed in advanced patients”. The authors should provide more justification, as these statements are of great importance when developing treatments for Covid-19.

The wording in “cumulative positive charge associated with the inserted HIV short sections”, together with “chimeric virus” in the abstract and in the main text could easily be understood to mean that SARS-CoV-2 is a fusion product between HIV and another coronavirus. The current consensus is that this (once) popular idea has no scientific merit. Even if this is not the authors’ intended interpretation, the phrases should be reworded for clarity.

---

## [Reviewer Report]

*Comments to Author*: Reviewer #1: Sørensen et al report a very interesting and timely bioinformatic analysis and study “Method of Action of Biovacc-19” in pre-clinical trial development. Biovacc-19 is a candidate vaccine for Covid-19 (SARS-COV-2) targeting the spike S1 protein that recognizes the ACE2 receptor for entry into cells for its mode of infection.Their strategy is to design the most effective vaccine that is both safe and able to combat SARS-COV-2. They focus on finding nonhuman-like (NHL) sequences 21.6% in the SARS-COV-2 spike S1 protein, rather than the 78.4% sequences already existing in humans. This strategy is sound and their approach may not only stimulate a good immune response, but also avoid triggering the human’s own immune system into attacking its own proteins, resulting in autoimmune diseases.

Sørensen et al base on their strategy on the analysis of others’ experience of developing HIV vaccine. They point out that despite years of effort, many-trial-and error studies, and a large amount of investment and clinical trials, an effective HIV vaccine still remains elusive.

To avoid taking the same path, they focus on nonhuman-like sequences. If their strategy works well for SARS-COV-2, their method could be generalized for many vaccines, both viral and nonviral pathogens.

Their analysis is sound and thorough. Through detailed alignment and attention to detail, they have uncovered some features that previous analyses missed. They noticed the pI differences between SARS-COV-1 and SARS-COV-2 and other similar coronaviruses. They found that the positively-charged amino acids lysine (K) and arginine (R) are in the specific locations that can form ionic bonds (salt bridges) to stabilize the S1 protein. They specifically pointed out the additional 3K and 3R may play a key role as an effective antigen to elicit body’s immune response.

They also emphasize that one crucial aspect for a successful vaccine is the choice of adjuvant which must be very carefully chosen as integral to the vaccine strategy from the beginning, not as an afterthought.Few people have paid such detailed attention explicitly. This reviewer believes they are correct in this regard from carefully reading the literature.

They provide a detailed Table 1 of nonhuman-like sequences found in SARS-COV-2. They systematically analyze the protein sequence of the SARS-COV-2 in 6-amino acid segments reviewing a total 155 segments. These segments are important for vaccine development as they demonstrated for Biovacc-19. They also compared various methods of generating vaccines and ranked the likelihood of success in Table 2, to provide some guidance for vaccine development.

There are some minor points to improve the manuscript:

1) They use many acronyms without defining them first and the reader quickly gets lost. This reviewer needed to check the literature and Wikipedia to understand these acronyms. For example: FcgR (p.2), BCG (p. 3), IMM-101 (p.3), CLEC4M/DC-SIGNR (p.4), CMV (p.4). It is common in immunology to use many acronyms.But the non-experts are quickly confused and have to check the literature to understand these acronyms before understanding their meaning.The late legendary Francis Crick advised this reviewer in 1991 “If you do not have to use acronyms, especially only two or three words, it is best to use the words, rather than acronyms”. This reviewer took this advice to heart.

2) The authors should provide a more detailed legend for Figure 1. They should also use arrows to pinpoint the key 3 K and 3 R in the SARS-COV-2 sequence for clarity. The authors should be explicit and show these amino acids in the well-refined X-ray crystal structure. Please see: https://pubmed.ncbi.nlm.nih.gov/32225175/

3) The authors should explain in the Table 1 legend how these peptide sequences are obtained; whether shifting two to five amino acids from the previous peptide sequence, or otherwise.It is best not to make the reader figure this out. The authors should be explicit.

4) Similarly, they should provide more detail for the legends of Figure 4, 5 and Figure 7.

5) There is a new paper Structural basis of receptor recognition by SARS-CoV-2 of a 2.68Å resolution of crystal structure just published in Nature, March 2020. It has more details than the CryoEM structure and show the exact location of interface between spike S1 and ACE2 receptor including several arginines.This reviewer suggests the authors use this structure to pinpoint where the 3 K and 3 R and other nonhuman-like sequences are located in the structure.

6) Please cite this reference:

Jian Shang et al Structural Basis of Receptor Recognition by SARS-CoV-2. 2020 May; 581(7807): 221-224. doi: 10.1038/s41586-020-2179-y. Epub 2020 Mar 30, 2020.

https://pubmed.ncbi.nlm.nih.gov/32225175/

Reviewer #3: The need for a vaccine is urgent as the Covid-19 pandemic continues to spread. “Biovacc-19”, from the company Immunor, is a peptide vaccine candidate targeting selected non-human-like epitopes of the SARS-CoV-2 spike protein. The manuscript “Biovacc-19: A Candidate Vaccine for Covid-19 (SARS-CoV-2) Developed from Analysis of its General Method of Action for Infectivity” by Sørensen, Susrud, and Dalgleish is based on the central idea that additional cationic amino acids on the SARS-Cov-2 spike protein (compared to other coronaviruses) increases its infectivity through electrostatic attraction to biomembranes and non-ACE2 cellular targets. Albeit an educated guess, in the absence of verification experiments, such a mechanism of action is still only a guess. Sørensen et al. should clearly state that the proposed mechanism of action is only based on observing the protein sequence and mapping charged amino acids on a 3D model of the spike protein.

Nevertheless, there are no peptide vaccines among the competitors currently in advanced clinical trials. Because Biovacc-19 represents an innovative and very different vaccine development strategy, I recommend the manuscript from Sorensen et al. to be published in QRB Discovery with minor revisions. Especially the selection for non-human-epitopes is a major laudable step in the development process of new vaccines. The manuscript is written less like a scientific report, and more as a justification for a product from Immunor already undergoing initial tests. However, the format of QRB Discovery should be able to accomodate such manuscripts.

The reviewer randomly found out that the section “Coutard et al furthermore state that conversely, the highly pathogenic … Hong Kong 1997 outbreak.” (after the word ‘that’) is verbatim from Coutard et al., but the length of the quote and the lack of quotation marks increases the risk of confusion. All verbatim citations in the manuscript should be clearly marked as such.

It is somewhat unclear if the authors imagine that the arginine rich insert 6 “RRAR” is membrane penetrating, or only acts as a membrane anchor on the surface. Figure 3 shows the action of a cell penetrating peptide, but they also write later “but not acting in the same way as a typical cell-penetration-peptide”. Does “RRAR” stay on the surface of the membrane, or cross the membrane boundary, and how does the activity change upon cleavage at the adjacent furin site?

From a physical perspective, the total number of positively charged amino acids in the six inserts (compared to SARS) is still very limited: KR in insert 1, HKK in insert 2, HR in insert 3, none in insert 4, none in insert 5, and RRR in insert 6. The manuscript mentions the unspecific effect of 7 positive charges from inserts 1-5 spread all over the protein (the basis of the higher isoelectric point of the spike protein of SARS-CoV-2 compared to SARS-CoV), and also the specific effect from the three arginines concentrated in insert 6. The two are fundamentally different. Comparing the two, should the latter be more relevant and dominating? If insert 6 really acts as a cell penetrating peptide in some way, why is “RRAR” only reported as a furin cleavage site in the literature, when occurring in other viruses?

Key references are missing mainly to Thorén and Åmand who were first to explain the basic uptake mechanisms of peptides, including the electrostatic effects that the authors claim they exploit. Some references directly related to the current study: https://doi.org/10.1016/S0014-5793(00)02072-X and https://doi.org/10.1016/j.bbamem.2011.03.011

The two charged domains, in the two Cys-Cys loops, should be marked out in Fig 5 for clarity.

There is very little proof in the literature that “it is evident that early phase Covid-19 is binding to the bitter/sweet receptors which provide a perfect mechanism for spread”. The virus could alternatively attack nerve cells, and indrectly destroy the taste buds, without directly binding to the receptors. Since this is a mechanism on the molecular level, the authors should expand on this claim and add more justification.

Likewise, there is very little proof that “The virus is thereby killing off erythrocytes which would account for the hypoxia observed in advanced patients”. The authors should provide more justification, as these statements are of great importance when developing treatments for Covid-19.

The wording in “cumulative positive charge associated with the inserted HIV short sections”, together with “chimeric virus” in the abstract and in the main text could easily be understood to mean that SARS-CoV-2 is a fusion product between HIV and another coronavirus. The current consensus is that this (once) popular idea has no scientific merit. Even if this is not the authors“ intended interpretation, the phrases should be reworded for clarity.